# Co-Expression of Multiple *PAX* Genes in Renal Cell Carcinoma (RCC) and Correlation of High *PAX* Expression with Favorable Clinical Outcome in RCC Patients

**DOI:** 10.3390/ijms241411432

**Published:** 2023-07-14

**Authors:** Lei Li, Caiyun G. Li, Suzan N. Almomani, Sultana Mehbuba Hossain, Michael R. Eccles

**Affiliations:** 1Department of Pathology, Dunedin School of Medicine, University of Otago, Dunedin 9016, New Zealand; lile2757@student.otago.ac.nz (L.L.); suzan.almomani@otago.ac.nz (S.N.A.); mehbuba.hossain@postgrad.otago.ac.nz (S.M.H.); 2Department of Radiation Oncology, Stanford University School of Medicine, Stanford, CA 94305, USA; caiyun.grace.li@gmail.com; 3Maurice Wilkins Centre for Molecular Biodiscovery, Level 2, 3A Symonds Street, Auckland 1010, New Zealand

**Keywords:** renal cell carcinoma, kidney cancer, *PAX* gene expression, clinical outcomes

## Abstract

Renal cell carcinoma (RCC) is the most common form of kidney cancer, consisting of multiple distinct subtypes. RCC has the highest mortality rate amongst the urogenital cancers, with kidney renal clear cell carcinoma (KIRC), kidney renal papillary cell carcinoma (KIRP), and kidney chromophobe carcinoma (KICH) being the most common subtypes. The Paired-box (*PAX*) gene family encodes transcription factors, which orchestrate multiple processes in cell lineage determination during embryonic development and organogenesis. Several *PAX* genes have been shown to be expressed in RCC following its onset and progression. Here, we performed real-time quantitative polymerase chain reaction (RT-qPCR) analysis on a series of human RCC cell lines, revealing significant co-expression of *PAX2*, *PAX6*, and *PAX8*. Knockdown of *PAX2* or *PAX8* mRNA expression using RNA interference (RNAi) in the A498 RCC cell line resulted in inhibition of cell proliferation, which aligns with our previous research, although no reduction in cell proliferation was observed using a *PAX2* small interfering RNA (siRNA). We downloaded publicly available RNA-sequencing data and clinical histories of RCC patients from The Cancer Genome Atlas (TCGA) database. Based on the expression levels of *PAX2*, *PAX6*, and *PAX8*, RCC patients were categorized into two *PAX* expression subtypes, PAXClusterA and PAXClusterB, exhibiting significant differences in clinical characteristics. We found that the PAXClusterA expression subgroup was associated with favorable clinical outcomes and better overall survival. These findings provide novel insights into the association between *PAX* gene expression levels and clinical outcomes in RCC patients, potentially contributing to improved treatment strategies for RCC.

## 1. Introduction

Approximately 430,000 people worldwide were diagnosed with kidney cancer in 2020, and around 179,000 kidney cancer patients died worldwide in the same year [1,2]. Renal cell carcinoma (RCC) is the most common type of kidney malignancy, accounting for approximately 85% of kidney cancer cases. Based on histological characteristics, RCC may be categorized into fifteen different subtypes [3,4]. Among these subtypes, kidney renal clear cell carcinoma (KIRC), kidney renal papillary cell carcinoma (KIRP), and kidney chromophobe carcinoma (KICH) are the most common. KIRC, which is the most common subtype, accounts for 75–80% of RCC. The next most common type is the KIRP subtype, which is characterized by papillary or tubular-papillary structures and accounts for 15–20% of RCCs. The KICH subtype typically contains a mixture of eosinophilic and clear cells, with abundant cytoplasm and well-defined cell boundaries, accounting for approximately 5% of RCCs. Metastasis and recurrence are common in RCC patients, leading to poor clinical outcomes. However, the lack of sensitive biomarkers for RCC tumors contributes to the challenges in effective management [5,6].

The current treatment modalities for RCC mainly include surgery, chemotherapy, immunotherapy, and molecular targeted therapy. However, RCC poses significant challenges in terms of treatment efficacy, as tumor metastasis and recurrence are frequently observed. In a recent commentary by Turajlic et al. [2] focusing on the molecular pathological characteristics, patient outcomes, and potential treatment strategies of kidney cancer, it was proposed that KIRC tumors with low intra-tumoral heterogeneity (ITH) and a low fraction of the genome affected by somatic copy-number alterations (SCNA) often exhibit reduced metastatic potential and more favorable overall prognosis. In contrast, tumors exhibiting increased SCNA, even with the presence of low ITH, tend to have enhanced metastatic potential and poorer prognosis. The authors emphasized the significance of understanding the molecular pathology of RCC in guiding treatment decisions for patients [2].

The *PAX* family of developmental control genes is frequently dysregulated in RCC [7,8]. This gene family, consisting of *PAX1*-*PAX9*, encodes a group of nine transcription factors (PAX1-PAX9), which are expressed in various human cancers and which have been implicated in the onset of malignancies, such as alveolar rhabdomyosarcoma [9]. Approximately 95% of RCC patients exhibit high expression levels of the *PAX2* and *PAX8* genes, and the expression of *PAX* genes has been associated with metastasis [10,11]. In previous studies on RCC, we demonstrated that knockdown of *PAX2* enhances cisplatin-induced apoptosis in RCC cells, suggesting that targeting of *PAX2* could be a potential therapeutic approach [12]. Furthermore, novel inhibitors of *PAX2* have been shown to suppress cancer cell proliferation in vitro [13]. Doberstein et al. found that *PAX2* binds to the A Disintegrin and metalloproteinase domain-containing protein 10 (*ADAM10*) promoter and that inhibiting *PAX2* expression significantly reduces the expression levels of ADAM10 protein in RCC cells [14]. This leads to a more diffuse cellular phenotype, accompanied by the upregulation of Snail family transcriptional repressor 2 (Slug) expression and the loss of E-cadherin, thereby promoting the migration of RCC cells [15]. Treatment with Transforming growth factor Beta 1 (TGF-β1) promotes epithelial-mesenchyme transition (EMT) of RCC cells in vitro, inducing stem cell-like characteristics in the cells [15]. In our previous work, we demonstrated that TGF-β1 treatment inhibits *PAX2* promoter activity, thereby suppressing *PAX2* expression in RCC cells [16].

In KIRC, the activation of a transcriptional program mediated by hypoxia-inducible factor alpha (HIFA) occurs due to *VHL* mutations, leading to the loss of von Hippel–Lindau (VHL) protein function. This, combined with the hypoxic conditions in the local tumor tissue microenvironment, has been shown to induce re-expression of *PAX2* [17]. The loss of VHL is observed in approximately 90% of KIRCs and leads to stabilization of HIF2A. Recent studies by Patel et al. have shown that HIF2A is preferentially recruited to transcriptional enhancers in chromatin where PAX8 is bound. This includes a pro-tumorigenic enhancer for cyclin D1 (*CCND1*), which is regulated by both PAX8 and HIF2A [18]. The interaction between PAX8 and HIF2A promotes the activation of several other oncogenes in KIRC. Therefore, targeting *PAX8* may offer a promising therapeutic approach for the treatment of RCC [19,20].

The above studies suggest that *PAX* genes play an important role in RCC. Consequently, we aimed to examine the relationship between *PAX* gene expression and clinical outcomes in RCC patients. Specifically, we investigated whether altered expression levels of *PAX* could be associated with poor clinical features such as invasion and metastasis in patients with advanced RCC.

In this study, we conducted a comprehensive analysis of publicly available renal cell carcinoma (RCC) data from The Cancer Genome Atlas (TCGA) to explore the significance of the *PAX* gene family in RCC (Figure 1). Additionally, we performed RT-qPCR analysis to assess the expression levels of *PAX* genes in various RCC cell lines. Using siRNAs, we selectively knocked down *PAX2* or *PAX8* genes in RCC cells and evaluated the effects on cell proliferation. Our results demonstrated that *PAX8* knockdown, but not *PAX2* knockdown, suppressed RCC cell proliferation. Furthermore, we classified RCC tumors into two distinct expression subgroups, namely PAXClusterA and PAXClusterB, based on the collective expression levels of *PAX* genes (*PAX2*, *PAX6*, and *PAX8*). We examined the correlation between these subgroups and the clinical characteristics of RCC patients. Our findings provide valuable insights into the potential implications of *PAX* gene expression in the treatment outcomes of RCC patients. Moreover, this study serves as a foundation for future investigations exploring the potential of *PAX* gene expression as a prognostic indicator or biomarker for treatment response in RCC.

## 2. Results

### 2.1. Clinical Characteristics of RCC Patients in the TCGA Cohort

In total, RNA-sequencing data and detailed clinical prognostic information from 883 patients represented in the TCGA database were incorporated in the present study. The patients included 288 females and 595 males with an age range of 17–90 years and a median age of 58 years. Among the 883 patients, 52.55% of the patients had stage I; 12.11% had stage II; 21.29% had stage III; and 11.66% had stage IV RCC. A summary of the clinical information including age at diagnosis, gender, ethnicity, and pathologic stage (T, N or M) is presented in Table 1.

### 2.2. Multiple PAX Genes Are Expressed in Clear Cell (KIRC), Papillary (KIRP), and Chromophobe (KICH) Renal Cell Carcinoma

In line with the workflow outlined in Figure 1, our initial analysis focused on examining the mRNA expression levels of *PAX* genes in three common subtypes of RCC (Figure 2; Appendix A), KICH, KIRC, and KIRP, using the publicly available TCGA dataset. Patients with incomplete clinical information were excluded from the study.

The mRNA expression levels of *PAX2*, *PAX6*, and *PAX8* were significantly elevated in the three common subtypes of RCC, as compared to the other members of the *PAX* family (Figure 2). This difference in expression was observed in contrast to the levels of *PAX1*, *PAX3*, *PAX4*, *PAX5*, *PAX7*, and *PAX9* (Appendix A).

### 2.3. Analysis of PAX Gene Expression Levels in Human RCC Cell Lines

To validate the expression patterns of *PAX* genes observed in the TCGA RCC dataset (Figure 2), we conducted RT-qPCR analysis on RCC cell lines (Figure 3). Relative expression levels of the reference genes are shown in Appendix A. The relative expression levels of *PAX2*, *PAX3*, *PAX5*, *PAX6*, *PAX8*, and *PAX9* were assessed in RCC cell lines, compared to a calibrator cell line, which exhibited the highest expression level (as designated by an asterisk in Figure 3). In agreement with the TCGA data analysis, *PAX2* showed relatively high expression levels in most RCC cell lines, with *PAX2* and *PAX8* exhibiting higher expression than *PAX6*. *PAX9* was also expressed at relatively low levels in seven out of nine RCC cell lines, while the expression of *PAX1*, *PAX4*, and *PAX7* was either very low or undetectable in RCC cell lines.

### 2.4. Effects of PAX2 or PAX8 Knockdown on A498 Cell Proliferation

Previous studies have shown that downregulation of *PAX* gene expression in cancer cells leads to reduced cell proliferation and induction of apoptosis [21,22,23]. In our earlier investigations [20], we provided evidence indicating that depletion of *PAX8* in RCC cell lines led to growth inhibition and initiation of senescence. In the present study, we utilized siRNAs to specifically target and suppress the expression levels of either *PAX2* or *PAX8* in A498 cells. Subsequently, we examined the impact of *PAX2* or *PAX8* depletion on A498 cell proliferation by employing the MTT colorimetric cell metabolic activity assay.

Cell proliferation in A498 RCC cells was monitored for six days following knockdown of *PAX2* or *PAX8*. Since *PAX6* was expressed at very low levels in the A498 cell line, *PAX6* knockdown was not performed. Successful knockdowns of PAX2 and PAX8 proteins were confirmed by Western blot analysis (Figure 4A) and RT-qPCR (Appendix A) at multiple timepoints. The RT-qPCR results demonstrated a significant decrease in mRNA expression levels of *PAX2* or *PAX8* in the A498 cell line after 72 h of knockdown, with expression levels being reduced to approximately 20% of the untreated (UN) group (Appendix A). The Western blot results further demonstrated significant inhibition of protein expression levels for both PAX2 and PAX8 at 96 h and 144 h after knockdown (Figure 4A).

In line with our previous data [20], the *PAX8* siRNA-treated samples exhibited a significant decrease in proliferation at 96 h (4 days) post-treatment compared to untreated (UN) or control siRNA-treated (SN, siControl) samples (Figure 4B). Moreover, two different siRNAs targeting *PAX8* (labeled S8 and A8) resulted in a significant reduction in cell number (proliferation) at 96 h (Figure 4C), which corresponded to the loss of PAX8 protein expression at these time points in the A498 cell line (Figure 4A). In contrast, despite siRNA knockdown of PAX2 protein expression at 96 h and 144 h (S2), the knockdown of *PAX2* did not significantly impact the proliferation of the A498 cell line compared to the negative control samples (Figure 4B).

### 2.5. Investigation of PAX Gene Expression Levels in Relation to Publicly Available Clinical Data from RCC Patients

To investigate the potential effects of *PAX* gene expression on clinical features in RCC patients, we combined the TCGA data for the three RCC subtypes (KIRC, KIRP, and KICH) into one analysis cohort. We compared the mRNA expression levels of *PAX1-9* in RCC samples and adjacent normal tissues (Figure 5A). Interestingly, *PAX2*, *PAX6*, and *PAX8* showed significantly higher mRNA expression levels in RCC samples compared to other members of the *PAX* gene family. However, these genes were significantly downregulated (*p* < 0.001) in the tumor samples compared to the adjacent normal kidney tissue (Figure 5A). Of note, the expression level of *PAX6* was relatively lower than *PAX2* and *PAX8* but still significantly above background.

### 2.6. Patients with Higher PAX2 and PAX8 mRNA Expression Exhibited Better Overall Survival in RCC

An analysis of *PAX2* and *PAX8* expression levels in relation to the survival of RCC patients revealed high expression levels of both *PAX2* (*p* < 0.001; Figure 5B) and PAX8 (*p* = 0.006; Figure 5C) in RCC which were associated with better overall patient survival. Specifically, we observed a positive correlation between *PAX2* expression and overall survival (OS) in patients from the KIRP (HR = 0.78), KIRC (HR = 0.63), and KICH (HR = 0.3) cohorts (Appendix A). Similarly, the expression of *PAX8* showed a positive correlation with OS in patients from the KIRP (HR = 0.59), KIRC (HR = 1.1), and KICH (HR = 0.11) subtypes. In contrast, *PAX6* expression did not show a significant association with overall patient survival. These results indicate that high expression of *PAX2* in KIRC patients and high expression of *PAX8* in KICH patients are associated with better prognosis (Appendix A).

### 2.7. Identification of PAXcluster A and PAXcluster B Subgroups in RCC Tumors

In both the RCC cell line investigations and the analysis of TCGA public data, it was observed that *PAX2*, *PAX6*, and *PAX8* were the most highly expressed *PAX* genes in RCC. High expression of these genes was generally associated with better OS in the KIRC, KIRP, and KICH RCC subtype cohorts (Appendix A). Using the unsupervised consensus clustering method “ConsensusClusterPlus” on the combined cohort of RCC patients (KIRC, KIRP, and KICH), we performed cluster analysis based on the expression profiles of *PAX2*, *PAX6*, and *PAX8* genes (Figure 6A). The analysis revealed that the *PAX* gene expression patterns of each cluster were highly significant by consensus matrix analysis and by principal component analysis (Appendix A), and from this analysis two different subtypes, which we called PAXcluster A and PAXcluster B, were identified. The results of a prognostic analysis showed that the PAXcluster A subtype, which has relatively high *PAX* expression, had a better survival advantage than the PAXcluster B subtype, which exhibited relatively lower *PAX* expression (Figure 6B). To explore biological processes between these two clusters, we performed a gene set variation enrichment analysis (GSVA) (Appendix A), which showed that the PAXcluster A was markedly enriched in pathways related to metabolism and repair, such as mammalian circadian rhythm, snare interactions in vesicular transport, histidine metabolism, base excision repair, and glycosylphosphatidylinositol (GPI) anchor biosynthesis. The PAXcluster B was mainly enriched in tumor-related pathways, such as alpha-linolenic acid metabolism, calcium signaling pathways, neuroactive ligand-receptor interactions, cardiac muscle contraction, extracellular matrix (ECM) receptor interactions, and aldosterone-regulated sodium reabsorption.

### 2.8. Clinical Features Are Associated with PAXcluster A and PAXcluster B Expression Subtypes in RCC Patients

The PAXcluster A subtype was observed significantly more frequently in female patients (*p* = 0.012). Furthermore, the PAXclusterA subtype was observed significantly more frequently in early-stage tumors: stage I and stage II (*p* < 0.001), grade 1 and grade 2 (*p* = 0.023), and T stage 1 (T1) and T stage 2 (T2) (*p* < 0.001). Our findings suggest that patients with RCCs of the PAXclusterA subtype may tend to have a better prognosis, and patients with RCCs of the PAXclusterB subtype may tend to have tumors that exhibit more advanced stage and grade. In contrast, there were significantly fewer N stage 0 (N0, *p* = 0.005) and M stage 0 (M0, *p* = 0.001) tumors exhibiting the PAXclusterA subtype (Figure 7). The results showed there was no relationship between PAXcluster and age (*p* = 0.696). A heatmap presenting the overall pattern of clinical characteristics is shown in Appendix A.

## 3. Discussion

The *PAX* gene family consists of highly conserved transcription factors that play crucial roles in embryonic development and organogenesis [24,25,26,27,28]. These genes are regulated in a temporal and spatial manner, and expression patterns of *PAX2* and *PAX8* are typically downregulated in the kidneys during fetal development as they undergo terminal differentiation [10]. Healthy postnatal tissues can also express *PAX* genes in a restricted fashion. For example, *PAX2* and *PAX8* have been shown to be re-expressed in postnatal kidneys in response to acute kidney injury, nephrotoxicity, and regenerative changes [25,28]. However, in various types of cancer, including RCC, *PAX* genes can be re-expressed in an aberrant manner, contributing to abnormal cell proliferation and cancer cell survival [7].

Here we examined the expression levels of *PAX* gene family members in RCC cell lines and in TCGA data derived from three RCC subtypes (KIRC, KIRP, and KICH). Analysis of the online data revealed that *PAX2* and *PAX8* were expressed at relatively high levels in RCC tumor tissues compared to other *PAX* genes, while *PAX6* was expressed at lower but still significant levels. We also found that *PAX2*, *PAX6*, and *PAX8* were expressed in most RCC cell lines, indicating their potential involvement in controlling cell proliferation, survival, and chemoresistance in RCCs [13].

To investigate the functional significance of *PAX2* and *PAX8* in RCC, knockdown experiments were performed in A498 cells. Interestingly, inhibition of *PAX8* expression significantly suppressed cell proliferation, while inhibition of *PAX2* had little to no effect. This finding is supported by previous studies demonstrating that PAX8 can activate genes involved in cell cycle regulation and metabolism, acting as a transcriptional coactivator in RCC cells [19]. Silencing *PAX8* has been shown to decrease RCC cell proliferation, suggesting it is potentially an oncogene in RCC. On the other hand, *PAX2* expression was found to be significantly reduced in high-grade RCC, particularly in the KIRC subtype, compared to low-grade RCC.

In the TCGA data analysis, we observed a correlation between the expression levels of *PAX2* and *PAX8* and the clinical outcomes of RCC patients. Higher expression of *PAX2* and *PAX8* was associated with better survival of RCC patients compared to those with lower expression levels. Specifically, higher *PAX2* expression was associated with improved overall survival in KIRC and KICH patients (although the latter was not statistically significant). Also, relatively higher *PAX8* expression was observed (although not statistically significant) in KICH patients with improved overall survival. These findings suggest that higher expression of *PAX2* and possibly of *PAX8* are associated with better clinical outcomes in RCC. The role of *PAX6* expression in RCC remains unclear, although its expression in normal brain tissue is associated with cell differentiation and migration [8,27].

By using the “ConsensusClusterPlus” R package, we stratified RCC patients into two subtypes based on the expression levels of *PAX2*, *PAX6*, and *PAX8*. Some clinical characteristics associated with RCC patients in the in silico analysis showed correlation with the PAXClusterA versus PAXClusterB subgroups. For instance, we found that PAXClusterA, which was associated with relatively higher *PAX* gene expression, correlated with better survival and prognosis of RCC patients, compared to PAXClusterB. PAXClusterA also tended to be associated with lower stages of RCC. Gene enrichment pathway analysis showed that the PAXClusterA subgroup was significantly associated with metabolism and repair-related pathways. Another important finding is that female RCC patients exhibit a higher proportion of PAXClusterA subtypes. This suggests that female patients could exhibit more favorable survival outcomes in association with PAXClusterA. Lee et al. showed that female RCC patients had a significantly higher survival rate than male patients [29]. Significant molecular differences may exist between male and female RCC patients, and therefore gender-specific RCC studies and personalized gender-specific therapies may be warranted.

Using co-transfection experiments, Schwarz et al. showed reciprocal inhibition of promoter/enhancer activity by the respective PAX2 and PAX6 proteins acting on each other’s gene promoters [30]. Therefore, very likely inter-regulatory relationships exist between different *PAX* genes that are expressed in the same RCC cells. *PAX2* and *PAX8* may play synergistic roles in gene regulation [31]. For instance, *PAX2* and *PAX8* are essential for initiating pro- and mesonephros development, and together they play a role in the expression of GATA binding protein 3 (*GATA3*) and initiating Hepatocyte nuclear factor 1 alpha, or beta (*HNF1b*a/b) expression [32,33]. Therefore, specific interactions between *PAX* genes likely occur in RCC, although further investigations will be needed to refine *PAX* gene interaction networks.

The re-expression of *PAX2* has been suggested to be required for renal tubular regeneration, proliferation, and repair [34]. Bleu et al. found that *PAX8* can activate metabolic genes through enhancer elements in renal cell carcinoma [19]. PAX8 has also been shown to transcriptionally activate (E2F transcription factor 1) *E2F1* expression, and hence the cell cycle, in RCC cells [20]. Moreover, *PAX2* and *PAX8* are potential oncogenes in RCC [35,36]. However, the full extent of mechanisms involved in their oncogenic effects remains unclear. Some studies have found that *PAX2* has both oncogenic and inhibitory effects on invasion during tumor development. In ovarian cancer, reduction of *PAX2* expression appears to be an early event of cancer clonal expansion, during which *PAX2* has inhibitory effects on tumor invasion and metastasis, potentially through interactions involving multiple pathways, including between Phosphatase and tensin homolog (*PTEN*), *PAX2*, and Tumor protein p53 (*TP53*) [37,38]. Transcription of *PAX2* was also shown to be dependent on *TP53* mutation status. *PAX2* was found to be directly transcriptionally activated by wildtype p53, but conversely *PAX2* transcription was inhibited by mutant p53 in murine oviduct epithelial cells [38]. In an in vitro invasion model using prostate cancer cells 22Rv1 and DU145, *PAX2* overexpression promoted prostate cancer cell invasion, which was associated with upregulated N-cadherin expression [39].

The expression of *PAX2* and *PAX8* in RCC may be associated with the maintenance of an epithelial phenotype during the epithelial-mesenchyme transition (EMT). In normal kidney development, PAX2 and PAX8 are necessary for the epithelial differentiation of renal precursor cells, and PAX2 promotes the trans-differentiation of mesenchymal cells into epithelial cells [28]. Thus, the downregulation of *PAX* gene expression in later stages of RCC may facilitate mesenchymal transition, which is linked to tumor cell invasion and metastasis. TGF-β signaling, elevated in advanced (stage III and IV) RCC tumors, has been shown to repress *PAX2* mRNA transcription through Mothers against decapentaplegic homolog 2, or 3 (SMAD2/3)-mediated mechanisms, potentially contributing to the reduced *PAX2* expression observed in advanced stages of RCC [16,40].

## 4. Materials and Methods

### 4.1. TCGA Database Patient Selection

A total of 883 RCC patients from the TCGA database were identified with KIRC, KIRP, or KICH. Of these, two patients did not have associated survival data; therefore, 881 RCC patients were included in the analyses of clinical features and outcomes, including patient overall survival.

### 4.2. Analysis Using Gene Expression Profiling Interactive Analyses {GEPIA) Online Tools

The GEPIA website (available at http://gepia.cancer-pku.cn/, accessed on 20 December 2022) is an online resource to analyze clinical data from TCGA database and tissue-specific expression patterns [41,42]. The relationship between the expression level of *PAX* genes and patient survival in different RCC subtypes was analyzed by GEPIA. Overall survival (OS) analysis was performed based on gene expression levels. The median gene expression level was the Group Cutoff between the high- and low-expression groups. A hypothesis test was performed using the Mantel–Cox test.

### 4.3. Unsupervised Clustering Based on PAX Genes

To further analyze the biological characteristics of the *PAX* genes in RCC patients, we performed unsupervised cluster analysis on the combined RCC patients to classify the RCC patients based on the expression level of the *PAX* genes for further analysis. We used the R software package “ConsensusClusterPlus” [43,44,45,46] to detect high consensus, optimal molecular subgroups based on *PAX2*, *PAX6*, and *PAX8* gene expression features. The clustering was performed by a K-means algorithm with Euclidean distance. The maximum cluster number was set to 9. The final cluster number was determined by the consensus matrix and the cluster consensus score (score > 0.8), and 50 iterations were used to assess the clustering stability.

### 4.4. Gene Set Variation Analysis (GSVA) and Functional Annotation

We identified the potential functional pathways of the PAX clusters using the “GSVA” R package [47]. The R package “clusterProfiler” was used to process biological-term classification and the enrichment analysis of gene clusters [48]. The gene set of “c2.cp.kegg.v7.2.symbols.gmt” obtained from the Molecular Signatures Database (MSigDB; https://www.gsea-msigdb.org, accessed on 10 January 2023) was used to run GSVA [49].

### 4.5. Cell Culture

All mammalian cell lines used in this study were grown in a humidified 37 °C incubator with 5% CO_2_. Cell line information, including its source and culture medium, is listed in Appendix A. To maintain a cell line in log-growth phase, the culture medium was refreshed every three days. Cells were sub-cultured when the confluence reached 90%.

### 4.6. Semi-Quantitative Reverse Transcription Real-Time PCR (RT-qPCR)

Experiments were performed as previously described [16]. For screening *PAX* gene expression in mammalian cell lines, 500 ng of total RNA from each cell line was used for generating cDNA in a 20 μL reaction. SuperScript III Reverse Transcriptase Kit (Invitrogen, Waltham, MA, USA) was used for synthesizing first-strand cDNA (primed with 0.5 mM dNTP mix and 250 ng random hexamers, Invitrogen, USA). Specific primers were used with the Platinum SYBR Green qRT-PCR SuperMix-UDG with ROX Kit (Invitrogen, USA) to amplify and detect target genes with three technical replicates for each cell line. To assess transcriptional expression, the fold change in the expression of *PAX* genes compared to three reference genes was employed (*PAX* gene expression levels were determined using the −∆∆Ct method). Gene expression data, with three technical replicates per knockdown, were normalized to the reference genes (Appendix A), as previously described [20]. The primer sequences used for RT-qPCR analysis are presented in Table 2.

### 4.7. siRNA Transfection

Gene silencing was achieved using gene-specific small interfering RNAs (siRNAs). The siRNAs used are listed in Appendix A and were transfected using “reverse transfection”. The transfection mix, which contained 10 nM (final concentration) siRNA, Lipofectamine RNAiMAX (Invitrogen, USA), and OPTI-MEM (Invitrogen, USA), was prepared according to the RNAiMAX instruction manual. Cells were harvested by trypsinization and counted using a hemocytometer. The siRNA transfection mix was added to a multi-well plate, followed by the drop-wise addition of the cell suspension, as described previously [20].

### 4.8. Western Blotting

Protein isolation and Western blots were performed as previously described [20]. Briefly, cells were trypsinized, washed in PBS, and lysed on ice for 30 min. Proteins were transferred from the SDS gel onto a nitrocellulose membrane (Hybond-C Extra) using a Mini Trans-Blot cell system (Bio-Rad Laboratories, Hercules, CA, USA). After transfer, the membrane was rinsed briefly in PBS with 0.1% Tween-20 (PBST) and transferred to blocking buffer for 1 h at room temperature. After blocking, the membrane was incubated in primary antibody diluted in blocking buffer (with the addition of 0.1% sodium azide) overnight at 4 °C. Next, the membrane was washed in PBST. The membrane was then transferred to the appropriate horseradish peroxidase (HRP)-conjugated secondary antibody (Sigma-Aldrich, Burlington, MA, USA) diluted in blocking buffer and incubated for 2 h at room temperature. Finally, protein detection was performed by incubating the membrane in freshly prepared SuperSignal West Pico Chemiluminescent Substrate. Antibody information and their optimized dilutions are listed in Appendix A.

### 4.9. Measurement of Cell Proliferation: The MTT Assay

For the MTT assay, cell transfections/manipulations were performed in duplicate in a 96-well plate format. Two controls were included: cells without siRNA transfection and cells transfected with the negative control siRNA. MTT assays were carried out using the Cell Proliferation Kit (Roche Applied Science, Penzberg, Bavaria, Germany) at selected time-points, following the manufacturer’s instructions. Absorbance of the final solution was measured at 570 nm using an Anthos ELISA plate reader, and values were expressed relative to the control samples.

### 4.10. Statistical Analysis

For RT-qPCR analysis, the qBase software (http://medgen.ugent.be/qbase/) (version qbase-windows-x64-v3.4) was used. For statistical analysis between two samples, the unpaired two-tailed t-test was used. For the comparison of categorical variables, we used the Chi-squared test. Bioinformatic analyses were performed using R software (version 4.1.2). We used the files that contain “fragments per kilobase per million” (FPKM) values. FPKM values were transformed into transcripts per kilobase million (TPM) values. Batch effects were removed from TPM expression data using the ComBat function from the “sva” R package [50]. The visualization of heatmaps and histograms was based on the “ggplot2” package [51]. Survival analysis was conducted based on R packages “survival” and “survminer” [52]. If not specified above, *p*-value < 0.05 was considered statistically significant for the results.

## 5. Conclusions

In conclusion, this study investigated the prognostic relevance of *PAX* gene expression levels and clinical outcomes in RCC. Higher expression of levels of *PAX2* and *PAX8* were associated with better overall survival in RCC patients. Stratification of patients based on *PAX* expression subtypes revealed that the subtype with relatively higher *PAX* expression (PAXCluster A) was associated with more favorable clinical features and improved survival outcomes. These findings contribute to our understanding of the role of *PAX* gene expression in RCC, and they may help guide future studies to improve treatment outcomes for RCC patients.

## Figures and Tables

**Figure 1 ijms-24-11432-f001:**
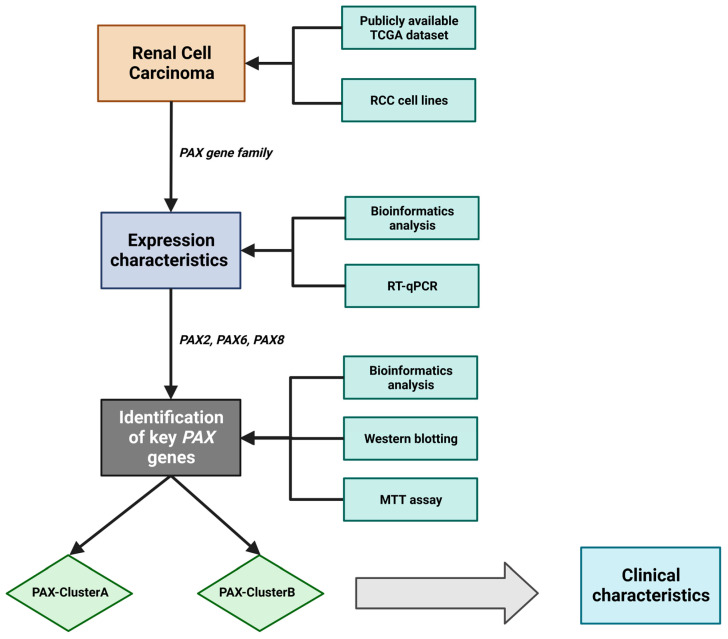
Flow diagram of the study. This study aims to investigate the expression levels and potential roles of the *PAX* gene family in RCC. RCC patients are classified based on the expression of key *PAX* genes, leading to the identification of two subgroups: PAXClusterA and PAXClusterB. Further, the study explores the differences in clinical outcomes between RCCs belonging to these subgroups.

**Figure 2 ijms-24-11432-f002:**
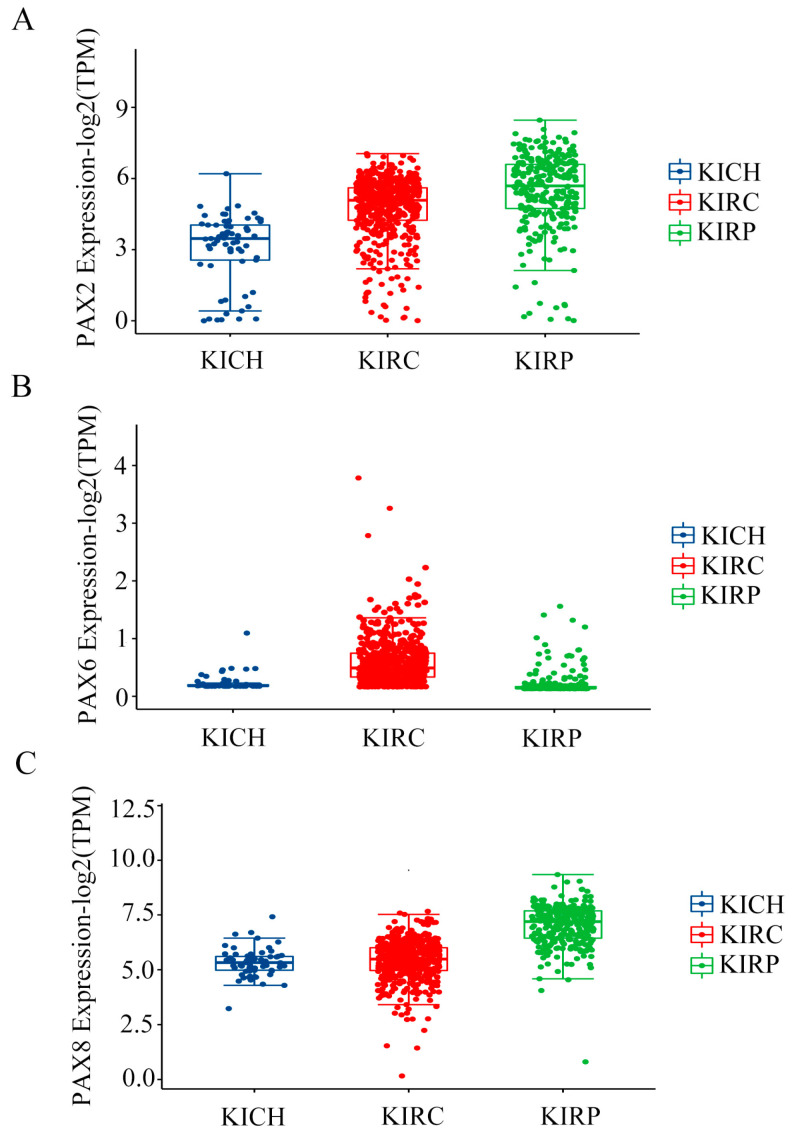
Analysis of *PAX* gene expression in RCC subtypes using the TCGA dataset. (**A**) Box plots depicting the expression levels of *PAX2* in three RCC subtypes. (**B**) Box plots displaying the expression of *PAX6* in three RCC subtypes. (**C**) Box plots displaying the expression levels of *PAX8* in three RCC subtypes. KICH (*n* = 65), KIRC (*n* = 530), KIRP (*n* = 288).

**Figure 3 ijms-24-11432-f003:**
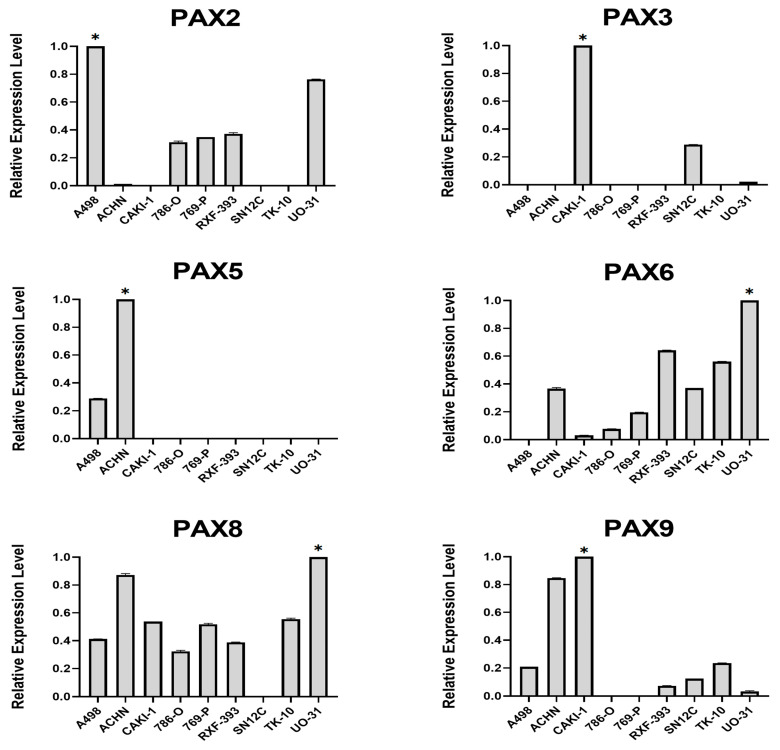
Analysis of *PAX* gene expression in RCC cell lines using RT-qPCR. The expression levels of multiple *PAX* genes were normalized to reference genes. The expression data are presented relative to the “calibrator” cell line, which exhibited the highest expression level (*). The gene expression profiles were determined using the −ΔΔCt method. RT-qPCR was performed with three biological replicates (*n* = 3).

**Figure 4 ijms-24-11432-f004:**
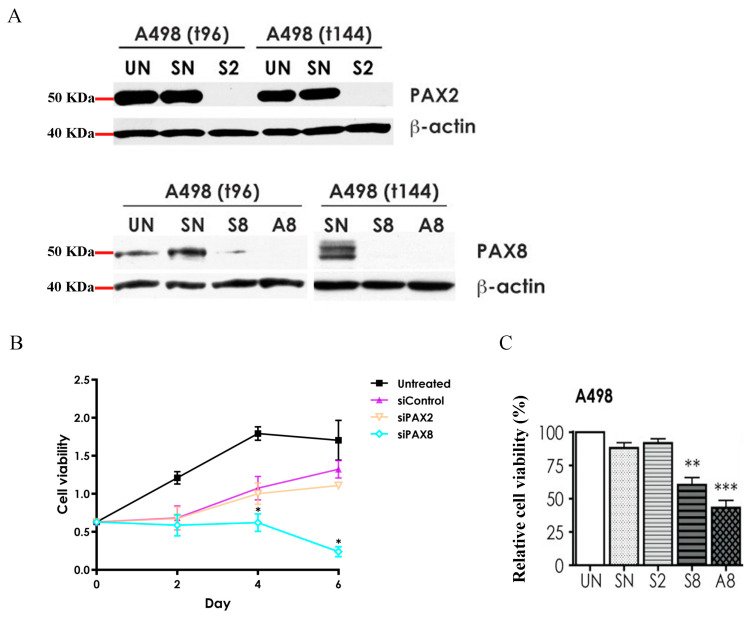
Effects of *PAX2* or *PAX8* knockdown on cell proliferation in A498 cell line. (**A**) Protein levels of PAX2 and PAX8 in A498 cell line were assessed using Western blots at various time points (as indicated) after transfecting with siRNA (S2, S8 or A8). Negative control samples were untreated (UN) and control siRNA-treated (SN) cells. (**B**) A498 cells were untreated (UN) or treated with SN, S2, or S8 on Day 0. Cell proliferation assays (MTT) were performed at the indicated time points (0, 2, 4, and 6 days post-treatment). (**C**) Cell proliferation was quantified (MTT) at 96 h post-treatment. *PAX8* knockdown was carried out using two different *PAX8* siRNAs (S8 and A8) to verify the proliferation reduction observed in A498 cells. siRNA concentration: 10 nm. *, *p* < 0.05; **, *p* < 0.01; and ***, *p* < 0.001 (one-way ANOVA, Tukey’s test).

**Figure 5 ijms-24-11432-f005:**
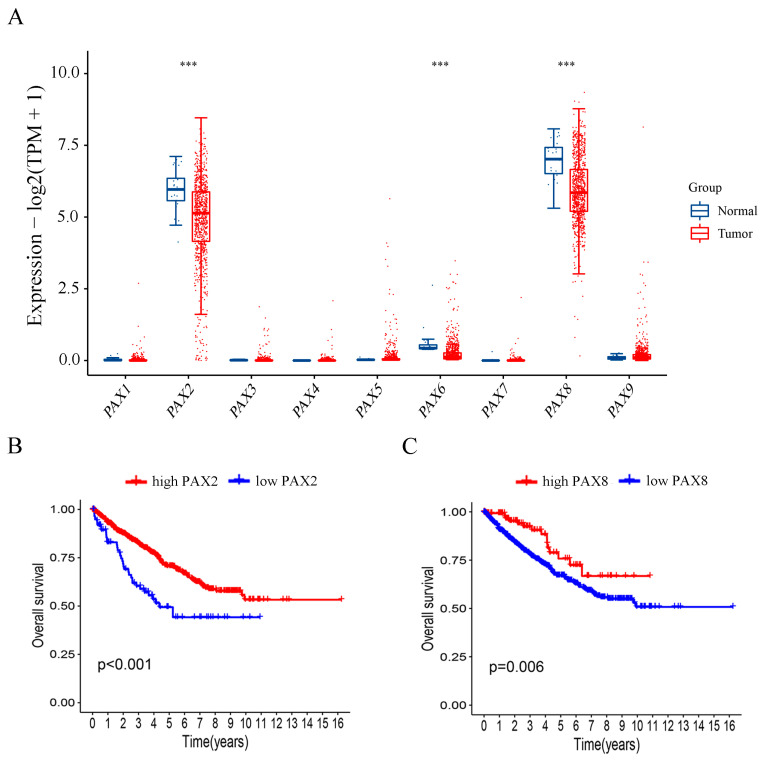
This figure illustrates the correlation between *PAX* gene expression and publicly available clinical data in RCC patients. (**A**) Gene expression levels of *PAX* genes in RCC compared to adjacent normal tissue; (**B**,**C**) Kaplan–Meier plots showing analysis of the relative level of *PAX2* (**B**) and *PAX8* (**C**) expression versus overall survival in RCC patients. RCC patient data were derived from three TCGA cohorts (TCGA-KIRC, TCGA-KIRP, and TCGA-KICH), while the data from a group of normal controls were derived from the normal adjacent/matched normal samples of RCC patients in TCGA. The analysis included a total of 881 patients. The significance level is indicated as *** *p* < 0.001.

**Figure 6 ijms-24-11432-f006:**
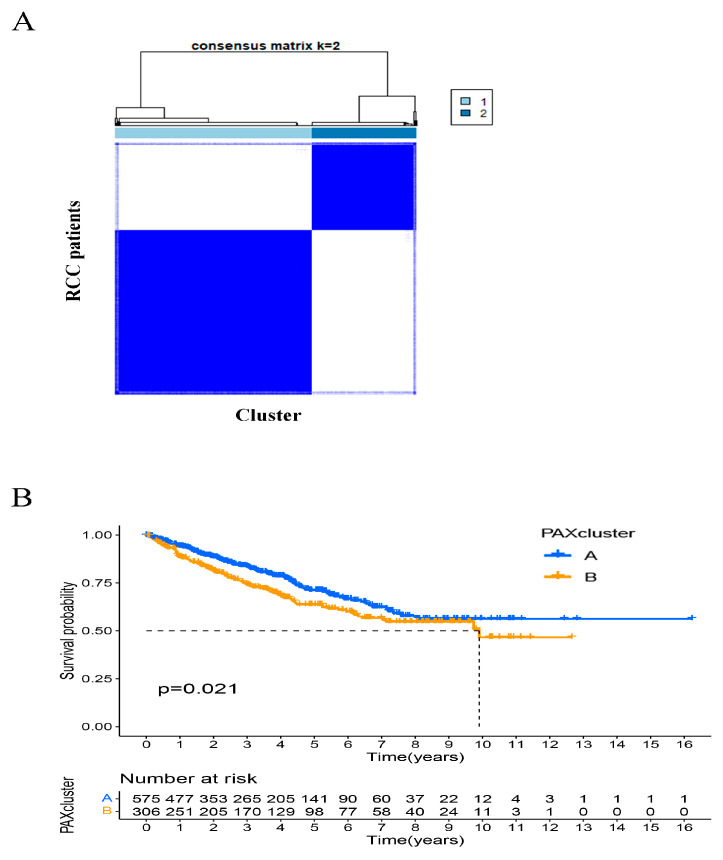
Unsupervised clustering and its association with RCC clinical outcome. (**A**) Heatmap representing the consensus matrix with a cluster count of two, which was determined using the minimal consensus score of >0.8. (**B**) Kaplan–Meier survival curve showing the relationship between *PAX* gene-related subtypes and overall survival. The PAXcluster A subtype has a better survival advantage than the PAXcluster B subtype. *n* = 881 patients.

**Figure 7 ijms-24-11432-f007:**
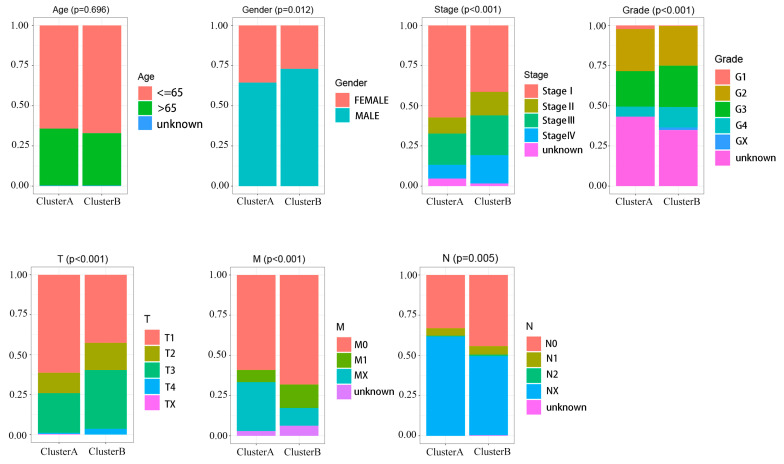
Identification of clinical features of PAXClusterA and PAXClusterB expression subtypes. The relationship between age, gender, stage, grade, T, M, N, and the PAXCluster subtypes, PAXClusterA (ClusterA) and PAXClusterB (Cluster B). The TNM system of cancer staging reflects the extent of primary tumor growth (T), the nodal status for metastasis (N), and the metastasis to distant organs (M). Statistical analysis was carried out using the Chi-squared test. *n* = 881 patients.

**Table 1 ijms-24-11432-t001:** TGCA-Patient Clinical Information.

	Clinical Features	KICH	KIRC	KIRP
Status	Alive	55	357	244
	Dead	10	173	44
Age	Mean (SD)	51.9 (14.1)	60.6 (12.1)	61.6 (11.9)
	Median [MIN, MAX]	50 [17, 86]	61 [26, 90]	61 [28, 88]
Gender	Female	26	186	76
	Male	39	344	212
Race	Asian	2	8	6
	Black	4	56	60
	White	57	459	205
	American Indian		2	
pT_stage	T1	20	271	199
	T2	25	69	36
	T3	18	179	47
	T4	2	11	2
	TX			4
pN_stage	N0	39	239	143
	N1	3	16	24
	N2	2		3
	NX	21	275	118
pM_stage	M0	50	440	205
	M1	2	80	9
	MX	13	10	74
pTNM_stage	I	20	265	179
	II	25	57	25
	III	14	123	51
	IV	6	82	15

**Table 2 ijms-24-11432-t002:** Gene-specific primers used in this study.

Target Gene	Forward Primer	Reverse Primer
*PAX1*	ACCCCCGCAGTGAATGG	TGTACACGCCGTGCTGGTT
*PAX2*	CCTGGCCACACCATTGTTC	TCACGTTTCCTCTTCTCACCAT
*PAX3*	ACGCGGTCTGTGATCGAAACA	TCTCGCTTTCCTCTGCCTCCTT
*PAX4*	CAGAGGCACTGGAGAAAGAGTTC	CCATTTGGCTCTTCTGTTGGA
*PAX5*	GTGCCTGGGAGTGAGTTTTCC	GGCGGCAGCGCTATAATAGT
*PAX6*	GAGGCTCAAATGCGACTTCAG	TGCTAGTCTTTCTCGGGCAAA
*PAX7*	GGAAGAAAGAGGAGGAGGATGAG	CCAGCCGGTTCCCTTTGT
*PAX8*	TGAGGGCGTCTGTGACAATG	CGGGACTCAGGGACTTGGT
*PAX9*	AGTACGGTCAGGCACCAAATG	ATAACCAGAAGGAGCAGCACTGTAG

## Data Availability

Datasets that were analyzed during this study are available in TCGA (https://portal.gdc.cancer.gov/, accessed on 20 December 2022).

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
