# Peer review of "Co-Expression of Multiple PAX Genes in Renal Cell Carcinoma (RCC) and Correlation of High PAX Expression with Favorable Clinical Outcome in RCC Patients"

_ijms, 2023, doi:10.3390/ijms241411432_

Round 1
Reviewer 1 Report (New Reviewer)
In this manuscript entitled "Co-expression of multiple PAX genes in renal cell carcinoma (RCC), and correlation of high PAX expression with favorable clinical outcome in RCC patients", the authors investigated the role of PAX genes family in the onset and progression of RCC. An interesting correlation between PAX gene expression and overall survival has been observed. However, there are some major concerns that should be addressed for the publication of this manuscript in IJMS.
1) Figure 3: the graphs of figure 3 lack the standard deviation. The experiment should be performed on at least 3 biological replicates. Moreover, I sincerely have some doubt about the analysis of the real time PCR. Why did the authors use three different housekeeping genes (PPIB, YWHAZ, HPRT1) for the analysis of real time? Are he graphs of figure 3 the sum of three different analysis? The authors should specify the reason of that.
2) Figure 4A: the western blot images show the silencing efficacy of PAX2 and PAX8 until day 6 post-silencing. However, the image is incomplete as it lacks t48 for PAX8 samples. Moreover, the authors should also provide the silencing effect on the mRNA of the respective genes. The concentration of the respective siRNA used (10nM, as indicated in materials and methods) should also be included in the caption of the figure or in the text.
Minor concerns:
1) Line 165-169: the authors stated that “When compared to untreated (UN), or control siRNA-treated (SN, siControl), the PAX8 siRNA-treated (S8, A8) samples exhibited significant repression of proliferation at 96 hours post-treatment. In contrast, cells treated with an siRNA targeting PAX2 (S2) demonstrated no significant effect on proliferation compared to negative control samples”. Is this referred to figure 4B? if yes, please clarify. However, graph in figure 4B does not show significance, in contrast to what the authors have claimed. Please edit the text.
2) Line 171-172: the authors stated “We did not observe any visible consequences of PAX siRNA treatment on cell viability in A498 cells.” Is this referred to figure 4D, bottom? If yes, please clarify. However, in contrast to what the authors have claimed, a significant reduction of cell viability of ~50% was observed with A8 siRNA. Please edit the text in accordance to what the figure show. Moreover, figure 4D upper is not mentioned in the text. Please add it.
3) Line 214-216: the authors stated “Furthermore, high expression of PAX8 in KICH patients was associated with a better prognosis (Supplementary Figure 5)".
Authors should note that also high PAX2 is associated with a better overall survival. please edit the text.
4) Line 408: the supplementary table 4 is missing in the supplementary file.
5) Figure 4C is not mentioned in the text. However, I sincerely believe it could be removed, as it does not add any important information.
6) Line 319: please edit RAX2 in PAX2
Overall, an improvement of data presentation/discussion and English style is necessary.
The English style does not sound very scientific throughout the text. Improvements of both language and data presentation are necessary for publication in IJMS.
Author Response
Comments and Suggestions for Authors
In this manuscript entitled "Co-expression of multiple PAX genes in renal cell carcinoma (RCC), and correlation of high PAX expression with favorable clinical outcome in RCC patients", the authors investigated the role of PAX genes family in the onset and progression of RCC. An interesting correlation between PAX gene expression and overall survival has been observed. However, there are some major concerns that should be addressed for the publication of this manuscript in IJMS.
1) Figure 3: the graphs of figure 3 lack the standard deviation. The experiment should be performed on at least 3 biological replicates. Moreover, I sincerely have some doubt about the analysis of the real time PCR. Why did the authors use three different housekeeping genes (PPIB, YWHAZ, HPRT1) for the analysis of real time? Are he graphs of figure 3 the sum of three different analysis? The authors should specify the reason of that.
R: Thank you for your comment. We have revised Figure 3 with the addition of standard deviations. Regarding three housekeeping genes, reference genes (also referred to as housekeeping genes) are frequently used to normalise mRNA levels between different samples. Indirect consequences of gene knockdown (for example, if any of the reference genes were by chance regulated by PAX transcription factor) the expression level of the reference gene might as a result vary in the cells in response to the knockdown, or among different tissues (PMCID: PMC1609175). Therefore, according to geNorm (https://genorm.cmgg.be/) up to 5 reference genes are initially chosen, which are then examined for their variability in gene expression in the conditions employed. The three least variable reference genes are then chosen as the reference genes for use in the cells under the conditions of interest. Therefore, we analysed initially 5 reference genes, and using geNorm, we chose the three least variable and most stable reference genes for normalizing. Following this, the expression of each target gene was normalized to the expression of the reference genes. Relative gene expression was then calculated using the qBase software (http://genomebiology.biomedcentral.com/articles/10.1186/gb-2002-3-7-research0034, https://doi.org/10.1186/gb-2002-3-7-research0034).
2) Figure 4A: the western blot images show the silencing efficacy of PAX2 and PAX8 until day 6 post-silencing. However, the image is incomplete as it lacks t48 for PAX8 samples. Moreover, the authors should also provide the silencing effect on the mRNA of the respective genes. The concentration of the respective siRNA used (10nM, as indicated in materials and methods) should also be included in the caption of the figure or in the text.
R: Thanks for your comment. Although we originally showed t48 for PAX2 in Figure 4A, we have now removed the Western blot at t48, since the functional effects of PAX siRNA knockdown in A498 cells were not evident at t48. Because the functional effects of PAX8 siRNA knockdown were evident at t96 and t144, Figure 4A now shows Western blot data for PAX2 and PAX8 protein levels at 96 hours and 144 hours, which correspond to functional effects on cell proliferation at 96 hours and 144 hours in Figures 4B and 4C.
We have now included Supplementary Figure 3 in the Supplementary Figures, which provides verification of successful PAX2 or PAX8 knockdown at the mRNA level based on the results of RT-qPCR.
More detailed description of the PAX gene silencing has been incorporated into the Results section (pages 5-6, lines 150-175), and the siRNA concentration has been added to the legend of Figure 4.
Minor concerns:
1) Line 165-169: the authors stated that “When compared to untreated (UN), or control siRNA-treated (SN, siControl), the PAX8 siRNA-treated (S8, A8) samples exhibited significant repression of proliferation at 96 hours post-treatment. In contrast, cells treated with an siRNA targeting PAX2 (S2) demonstrated no significant effect on proliferation compared to negative control samples”. Is this referred to figure 4B? if yes, please clarify. However, graph in figure 4B does not show significance, in contrast to what the authors have claimed. Please edit the text.
R: Thank you for your comment. We have revised the text to clarify the description of the results (now described in lines 167-175). Based on the observations from Figure 4B, it is evident that the knockdown of PAX2 does not exert a significant effect on the proliferation of A498 cells (Day 0-6). Conversely, the knockdown of PAX8 leads to a reduction in the proliferation of A498 cells, which becomes noticeable starting from day 4 (96 hours). To enhance the clarity of our results, we have made modifications to Figure 4B by highlighting specific lines using distinct colors and indicating the significant differences*.
2) Line 171-172: the authors stated “We did not observe any visible consequences of PAX siRNA treatment on cell viability in A498 cells.” Is this referred to figure 4D, bottom? If yes, please clarify. However, in contrast to what the authors have claimed, a significant reduction of cell viability of ~50% was observed with A8 siRNA. Please edit the text in accordance to what the figure show. Moreover, figure 4D upper is not mentioned in the text. Please add it.
R: Thanks for your comment. We have made modifications to this figure and additions to the text (now described in lines 173-175). We have modified what was the original Figure 4D, and for improved clarity of presentation, we have removed the upper portion of the image in Figure 4D. The lower part is now shown as Figure 4C in the revised manuscript, which is referred to in the main text (line 226). We discuss in the main text (lines 173-175) that the knockdown of PAX2 does not have a significant effect on the proliferation of A498 cells. However, the knockdown of PAX8 demonstrates a significant inhibition of the proliferation rate of A498 cells.
3) Line 214-216: the authors stated “Furthermore, high expression of PAX8 in KICH patients was associated with a better prognosis (Supplementary Figure 5)".
Authors should note that also high PAX2 is associated with a better overall survival. please edit the text.
R: Thanks for your comment. In the case of KICH patients, the correlation between the expression level of PAX2 and prognosis does not reach statistical significance (P > 0.05), as is demonstrated in our new Supplementary Figure 6. This is discussed in lines 302-309.
4) Line 408: the supplementary table 4 is missing in the supplementary file.
R: Thanks for your comment. We have now added Supplementary Table 4.
5) Figure 4C is not mentioned in the text. However, I sincerely believe it could be removed, as it does not add any important information.
R: Thanks for your suggestion. We agree with your suggestion, and we have removed what was the original Figure 4C accordingly.
6) Line 319: please edit RAX2 in PAX2
R: Thanks for your comment. We have corrected it.
Overall, an improvement of data presentation/discussion and English style is necessary.
R: Thanks for your comment. We have edited the manuscript for data presentation and discussion, and for English language usage.
Comments on the Quality of English Language
The English style does not sound very scientific throughout the text. Improvements of both language and data presentation are necessary for publication in IJMS.
R: Thanks for your comment. We have edited the manuscript for English language usage.
Reviewer 2 Report (New Reviewer)
This is an interesting article, which may have an interest for the readers. However, this work raises some comments which must be addressed before considering the manuscript for publication.
Major comments
What is intriguing is while the authors show that higher PAX2 and PAX8 mRNA expression correlates to better overall survival in RCC, PAX8 appears to be critical for RCC cells proliferation. The authors do not provide any explanation/comment.
Moreover, arguing that PAX8 is critical for cell proliferation based on investigation of ONE cell line (at the end of the day, one cell line translate to one patient) does not follow the scientific rigor. The authors must show data from proliferation assay for at least few more cell lines. Also, it will very interesting to do these experiments comparing the effect of siRNA PAX8 on proliferation on VHL-wild type vs. VHL-mutated cells.
Showing that tPAX cluster A subtype expression was significantly higher in female requires a deep discussion even further investigation. This is an important observation
Minor comments:
Many paragraphs in the Introduction section fit much better in the Discussion section.
It is not clear how many repeats were done for the in vitro experiments.
Figures 4B-D are not cited in the main text. Also, I am not sure why the authors show images only for untreated cells.
What is the source of error bars in Figure 4B, average of replicates within the same experiment or average of multiple experiments?
Were mycoplasma cell culture test and cell authenticity assay performed?
Minor editing!
Author Response
Comments and Suggestions for Authors
This is an interesting article, which may have an interest for the readers. However, this work raises some comments which must be addressed before considering the manuscript for publication.
Major comments
What is intriguing is while the authors show that higher PAX2 and PAX8 mRNA expression correlates to better overall survival in RCC, PAX8 appears to be critical for RCC cells proliferation. The authors do not provide any explanation/comment.
R: Thanks for your comment. We have added an explanation to the Discussion (see lines 334-359, and particularly in lines 334-339).
Moreover, arguing that PAX8 is critical for cell proliferation based on investigation of ONE cell line (at the end of the day, one cell line translate to one patient) does not follow the scientific rigor. The authors must show data from proliferation assay for at least few more cell lines. Also, it will very interesting to do these experiments comparing the effect of siRNA PAX8 on proliferation on VHL-wild type vs. VHL-mutated cells.
R: Thanks for your comment. In our previously published paper [PMID: 21602887], we showed that siRNA-mediated PAX8 knockdown inhibits tumor cell proliferation in several RCC cell lines (A498, 786-O, CAKI-1). We now show in our current manuscript (using A498 cells) that the experiments again confirm this effect that was seen in our previously published PAX8 knockdown experiments, and our current manuscript refers to, and cites our previously published paper (lines 167-172; lines 336-337). Therefore, as our current work is a confirmation of the previous work in A498 cells, plus it extends the analysis in A498 cells to PAX2 knockdown, and given that multiple cell lines were used in our previous work, we feel it would be redundant to repeat the analysis with multiple cell lines in the current manuscript.
As the reviewer indicates, VHL is as a commonly mutated gene in RCC, particularly in KIRC. To date, we have investigated the effect of siRNA knockdown of PAX8 on the proliferation of KIRC cell lines (A498, 786-O; see data also presented in our previously published paper, PMID: 21602887). However, we haven’t confirmed whether these KIRC cell lines are VHL mutant cell lines. We have not investigated the effect of knockdown of PAX8 in KIRP cell lines, such as in ACHN cells (presumably VHL wild-type), although this would be an interesting experiment for future research. In our view, analysis of the role of PAX8 in VHL mutant and wildtype cell lines falls outside our present article’s focus.
Showing that tPAX cluster A subtype expression was significantly higher in female requires a deep discussion even further investigation. This is an important observation
R: Thanks for your comment. We have further discussed these findings in the Discussion section of the manuscript (lines 318-324, page 11).
Minor comments:
Many paragraphs in the Introduction section fit much better in the Discussion section.
R: Thanks for your comment. We have revised the Introduction section and have relocated some parts from the Introduction to the Discussion section (see lines 335-349, page 11-12).
It is not clear how many repeats were done for the in vitro experiments.
R: Thanks for your comment. We have added this information in the methods (lines 413-418, page 13).
Figures 4B-D are not cited in the main text. Also, I am not sure why the authors show images only for untreated cells.
R: Thanks for your comment. We have modified the description and interpretation of Figure 4 to provide a clearer explanation, and all panels of Figure 4 are now cited in the main text (see response to Reviewer #1). Furthermore, in accordance with the suggestion made by Reviewer #1, the original Figure 4C has been removed from the manuscript.
What is the source of error bars in Figure 4B, average of replicates within the same experiment or average of multiple experiments?
R: Thanks for your comment. The source of error bars in Figure 4B is the mean of replicates within the same experiment, with at least three replicates for each experiment to ensure the statistical robustness and reliability of the results.
Were mycoplasma cell culture test and cell authenticity assay performed?
R: Thanks for your comment. Yes, we performed mycoplasma cell culture testing and STR profiling as a cell authenticity assay on A498 cells, as part of our experimental procedures.
Comments on the Quality of English Language
Minor editing!
R: Thanks for your comment. We have edited the manuscript for English language usage.
Reviewer 3 Report (New Reviewer)
The authors present an interesting study examining the influence of paired-box (PAX) genes on cell phenotype and behaviour/s within the context of renal cell carcinoma. Briefly, the authors examine the expression profile of the PAX genes in a number of representative cell types commonly utilised in renal cell carcinoma studies. Within the family, PAX 2, 6, and 8 exhibited correlative expression patterns as compared to the others, prompting investigation of their role through knockout studies. In particular, PAX8 knockout ameliorated behaviours common of a cancer phenotype, with a reduction in cell proliferation observed. To close, the author compare their findings to open access clinical data, highlighting certain expression profiles of the PAX gene family are more clinically favourable than others, underscoring the in vitro findings from this article and others.
In reviewing the manuscript, I made a number of observations. The following should be addressed when preparing a suitable revision.
1. The writing is good overall but there are a few typos within the text. The authors should look at address these in any resubmission.
2. Can the authors clarify what is meant by a reverse transfection? More details on how this was carried out are required, and supporting data validating the KO of the gene should be included in the publication.
3. Details on the antibodies used are required. This includes but is not limited to the source, and the concentrations employed.
4. Were the primers MIQE guidelines compliant?
5. For the PCR analysis, should the method be delta delta Ct or simply delta Ct?
6. N-numbers for each study should be denoted in every figure and/or figure legend for clarity.
Apart from a few typos, the writing is of a good standard
Author Response
Comments and Suggestions for Authors
The authors present an interesting study examining the influence of paired-box (PAX) genes on cell phenotype and behaviour/s within the context of renal cell carcinoma. Briefly, the authors examine the expression profile of the PAX genes in a number of representative cell types commonly utilised in renal cell carcinoma studies. Within the family, PAX 2, 6, and 8 exhibited correlative expression patterns as compared to the others, prompting investigation of their role through knockout studies. In particular, PAX8 knockout ameliorated behaviours common of a cancer phenotype, with a reduction in cell proliferation observed. To close, the author compare their findings to open access clinical data, highlighting certain expression profiles of the PAX gene family are more clinically favourable than others, underscoring the in vitro findings from this article and others.
In reviewing the manuscript, I made a number of observations. The following should be addressed when preparing a suitable revision.
- The writing is good overall but there are a few typos within the text. The authors should look at address these in any resubmission.
R: Thanks for your comment. We have carefully reviewed the manuscript and corrected the typos.
- Can the authors clarify what is meant by a reverse transfection? More details on how this was carried out are required, and supporting data validating the KO of the gene should be included in the publication.
R: Thanks for your comment. Reverse transfection is a method of introducing nucleic acids, including small interfering RNA (siRNA), into cells. Unlike standard transfection, where cells are first seeded into the plate and then transfected with nucleic acids, in reverse transfection the cells are seeded directly onto siRNA-transfection complexes, allowing the nucleic acids to be immediately taken up by the cells, which reduces the duration of the transfection protocol by a day (a more detailed description is given in PMID: 18023808).
The specific steps are described in the methods (lines 423-430, page 14; “siRNA transfection”). Reverse transfection is a commonly used method, which we have also used in our previous publications (e.g. PMID: 21602887).
Successful knockdowns of PAX2 and PAX8 were confirmed by Western blot (Figure 4A) and also RT-qPCR (new Supplementary Figure 3) at multiple timepoints.
- Details on the antibodies used are required. This includes but is not limited to the source, and the concentrations employed.
R: Thanks for your comment. We have added this information in Supplementary Table 4.
- Were the primers MIQE guidelines compliant?
R: Thanks for your comment. To ensure adherence to the MIQE (Minimum Information for Publication of Quantitative Real-Time PCR Experiments) guidelines, this was used in designing the primers used in our study, which followed established guidelines, including considerations for specificity, efficiency, and other relevant parameters outlined in the MIQE guidelines. The sequences of the primers are shown in Table 2.
- For the PCR analysis, should the method be delta delta Ct or simply delta Ct?
R: Thanks for your comment. In this paper, we utilized the delta delta Ct (ΔΔCt) method for RT-qPCR analysis.
- N-numbers for each study should be denoted in every figure and/or figure legend for clarity.
R: Thanks for your comment. We have added the number of samples per experiment/analysis at the end of the figure legends of figure 1, figure 5, figure 6, and figure 7. For cell culture experiments (figure 3 and figure 4), we have indicated the number of experimental replicates, as indicated in the RT-qPCR methods on p13, lines 414 to 418.
Comments on the Quality of English Language
Apart from a few typos, the writing is of a good standard
R: Thanks for your comment. We have corrected all typos.
Round 2
Reviewer 1 Report (New Reviewer)
The present manuscript has been extensively improved compared to its first version. The required additional experiment have been performed and now the results are more consistent. The English style throughout the manuscript has improved and sounds more scientifically appropriate.
One last inacuracy:
Line 171: The authors stated that figure 4C showed reduction in proliferation at 96h and 144h. The caption of figure 4C instead refers only to 96h. Please be consistent.
Line 18: “using RNAi” is written twice in the same period. Remove the second one.
Reviewer 2 Report (New Reviewer)
I would like to congratulate the authors for their answers and overall work. From my perspective, the manuscript is suitable for publication now.
Reviewer 3 Report (New Reviewer)
The authors have addressed my concerns and the manuscript is improved in their responses to such
This manuscript is a resubmission of an earlier submission. The following is a list of the peer review reports and author responses from that submission.
Round 1
Reviewer 1 Report
The authors presented a brief analysis of the relationship between the PAX genes family and the survival of RCC patients, showing that upregulation of PAX2, PAX6, and PAX8 could be potentially beneficial for RCC patients. The current version needs certain modifications by addressing logical and statistical issues before its publication.
1. The authors stated that the frequencies of PAX2/6/8 were most common in RCC patients; however, in figs 2 and 3, supplements as well, they used expression levels instead. Frequency is a counting result in statistics by defining the number of certain "+" divided by the total number of "+" and "-". The results from these figures could only tell me that the expression levels of PAX2/6/8 are way higher than 0, while the others were statistically undetectable.
2. What is the relation of the siRNA experiments in section 2.2 with the main theme of the paper? It is quite strange that it has nothing to do with the subsequent analyses. Besides, where is "S6" in A498 cell line of Figure 4?
3. How to define PAX high and low in section 2.3? What is the difference between the PAX high and low from PAX high and low in section 2.4? Are they the same? If yes, I don't get the logic. If no, then, address the first sentence, please.
4. Detail the matrix in the supplementary figure 6? What are the row and column names?
5. Please describe more regarding 'ConsensusClusterPlus' in the materials and methods section.
6. Add survival analysis in the statistics section.
7. t-test should be used in figure 2. One-way ANOVA can be used but the authors should prove the two groups have equal variance.
8. How do the authors analyze figure 7? Proportional t-test could only apply to the first two graphs.
9. PAX6 is only marginally higher than 0 in the paper. Why not exclude the PAX6 result? Logically, we could see that in normal tissues PAX2 and PAX8 have higher expression. Then, higher expression of PAX2 and PAX8 might be beneficial. But how to interpret the PAX6?
Author Response
- The authors stated that the frequencies of PAX2/6/8 were most common in RCC patients; however, in figs 2 and 3, supplements as well, they used expression levels instead. Frequency is a counting result in statistics by defining the number of certain "+" divided by the total number of "+" and "-". The results from these figures could only tell me that the expression levels of PAX2/6/8 are way higher than 0, while the others were statistically undetectable.
We thank the reviewer for this comment, and we agree that the use of the word “frequency” was not correct in this context. Accordingly, we have changed our text in lines 78-81, and in lines 93-96, as well as lines 181-183 to indicate the levels of expression of PAX genes, rather than the frequency.
- What is the relation of the siRNA experiments in section 2.2 with the main theme of the paper? It is quite strange that it has nothing to do with the subsequent analyses. Besides, where is "S6" in A498 cell line of Figure 4?
In retrospect, we agree with the reviewer, that the siRNA experiments in section 2.2 do not easily fit with the main theme of the paper. Therefore, we have decided to remove the siRNA experiments from this paper, and accordingly Figure 4, and the associated text from section 2.2 have been deleted.
- How to define PAX high and low in section 2.3? What is the difference between the PAX high and low from PAX high and low in section 2.4? Are they the same? If yes, I don't get the logic. If no, then, address the first sentence, please.
The wording of “PAX high” and “PAX low” levels in section 2.3 was not intended to have the same meaning as PAXhigh and PAXlow in section 2.4. The text in section 2.3 refers to the high and low expression levels of a single PAX gene (such as PAX2 or PAX8). However, in contrast, in section 2.4, the combined levels of different PAX gene expression were calculated for a combined cohort of RCCs. Following this, combined values were calculated and depicted as individual clusters, which we have referred to as PAXcluster A, and PAXcluster B. Due to the wording “PAXhigh and “PAXlow” having caused confusion, we have therefore decided in section 2.4 not to use the words “PAXhigh” or “PAXlow”, and instead we have changed these to "PAXcluster A", and "PAXcluster B" in all cases. We have also changed the notations on the graphs in Figure 6 to “Cluster A”, and “Cluster B”, corresponding to PAXClusterA and PAXClusterB. We think making these changes will help reduce confusion.
- Detail the matrix in the supplementary figure 6? What are the row and column names?
The consensus matrix plot, previously indicated as being in supplementary figure 6, and which is now Figure 5A, shows the “cluster” on the X-axis (columns), denoting the cluster number (cluster 1 or cluster 2) in the columns. The blue square denotes these patients cluster together, and white sequres denote that these patients do not cluster together. The Y-axis (rows) indicates the RCC patients. On Figure 5A we have now indicated the labels on the rows and columns.
- Please describe more regarding 'ConsensusClusterPlus' in the materials and methods section.
We have added the following sentences in lines 291-297 under "Supervised clustering based on PAX genes" to better describe the 'ConsensusClusterPlus' in the methods section: “We used the R software package "ConsensusClusterPlus" [40-43] to detect high consensus, optimal molecular subgroups based on PAX2, PAX6 and PAX8 gene expression features. The clustering was performed by a K-means algorithm with Euclidean distance. The maximum cluster number was set to 9. The final cluster number was determined by the consensus matrix and the cluster consensus score (score >0.8), and 50 iterations were used to assess the clustering stability.”
- Add survival analysis in the statistics section.
We thank the reviewer for this. We have added the "survival analysis" to "Statistical Analysis" on lines 311-313.
- t-test should be used in figure 2. One-way ANOVA can be used but the authors should prove the two groups have equal variance.
We agree with the reviewer. Now we have revised Figure 2, and we no longer compare between tumour and adjacent normal tissue in Figure 2. Nevertheless, we compare between tumour and adjacent normal tissue in Figure 4, and we have used t-test in Figure 4.
- How do the authors analyze figure 7? Proportional t-test could only apply to the first two graphs.
We agree with the reviewer’s comments. The statistical analysis for this figure (which is now Figure 6) was carried out using Chi-squared test. The Chi-squared test has been shown to be useful for statistical analysis of tumour TNM data (Hu et al., 2015; doi: 10.3978/j.issn.2072-1439.2015.04.09).
- PAX6 is only marginally higher than 0 in the paper. Why not exclude the PAX6 result? Logically, we could see that in normal tissues PAX2 and PAX8 have higher expression. Then, higher expression of PAX2 and PAX8 might be beneficial. But how to interpret the PAX6?
We think that, even though PAX6 was not expressed at high levels in comparison to PAX2 and PAX8 in tumour tissues in the RCC patients, either in the TCGA data (in Figure 2), or in the RCC cell lines (in Figure 3), the PAX6 expression data were nevertheless significantly elevated in comparison to other PAX genes, such as PAX1. While the PAX6 transcripts were expressed at relatively low levels, they could be functionally significant, and transcript and corresponding protein levels are not necessarily concordant in tissues. We therefore think the PAX6 data is relevant to include in this paper.
Reviewer 2 Report
This study examined the expressions of genes from the PAX family in RCC and observed higher expressions of PAX2 and PAX8 in certain subtypes. They classified CCR samples based on PAX genes into 2 groups and observed clinical differences. Here are my questions:
1. How did the knockdown experiment help to identify key PAX genes? You used siRNA to target individual genes which showed decreased expression afterward. But what’s your point or discovery here? You may need to add more descriptions and discussions regarding Fig4.
2. In Fig2, significant differences were only seen in certain subtypes, but not all. Then why do you want to merge the subtypes in Figure 5? You said “to simplify these analyses”, however, this is not a solid justification. Were you able to see the impact of PAX2 and PAX6 on the survival of each subgroup?
3. Fig 3 is no “co-expression”. You may need to check the correlation of any two genes to claim “co-expression”.
Author Response
- How did the knockdown experiment help to identify key PAX genes? You used siRNA to target individual genes which showed decreased expression afterward. But what’s your point or discovery here? You may need to add more descriptions and discussions regarding Fig4.
As we have indicated to Reviewer #1, we have decided to remove the siRNA experiments from this paper, and accordingly Figure 4, and the associated text from section 2.2 have been deleted.
- In Fig2, significant differences were only seen in certain subtypes, but not all. Then why do you want to merge the subtypes in Figure 5? You said “to simplify these analyses”, however, this is not a solid justification. Were you able to see the impact of PAX2 and PAX6 on the survival of each subgroup?
We combined the survival data of the three RCC subtype patients so as to merge them into one RCC cohort to carry out the analysis in Figure 4 (which was previously Figure 5). This allowed us to do a high-PAX2 versus low-PAX2, and high-PAX8 versus low-PAX8 comparisons across all RCC patients at once. We did this because we observed significant impact of the high versus low PAX2 expression on the survival of patients with a KIRC subtype of RCC, and we also observed suggestive evidence (although it was not statistically significant) for an impact of PAX2 and PAX8 on the KICH subtype of RCC. Therefore, we thought that combining the RCC patients into one RCC cohort would be helpful for investigating the impact of PAX2 or PAX8 expression.
- Fig 3 is no “co-expression”. You may need to check the correlation of any two genes to claim “co-expression”.
Thank you for this suggestion. We have now changed “co-expression” in the description of Figure 3 to “expression”.
Reviewer 3 Report
Dear Authors,
This report concerns a study on the expression of selected PAX genes in RCC and links them with clinical outcome in RCC patients. The Authors performed a mixed analysis using publicly available data from CTGA and some experimental data from study of selected cancer-derived cell lines.
The novelty of the study comes from exploration of the expression of PAX genes as a prognostic factor in the clinical outcome of RCC.
Overall, I found the paper to be interesting.
Nevertheless, I have some major concerns and comments regarding the paper.
1. The Introduction is well-written and informative.
2. The Results section.
a/ First of all, the graphics are poor – in terms of quality, but also readability. They lack important information. E.g. Figure 2 A-C is missing information regarding gene names. Please provide the names of the genes on the Y-axis. Also, the description of the bars (e.g., “T”- tumor and “NT” non-tumor should be added).
In case of Fig. 3 – please provide the names of the cell lines (below the X-axis) on each graph.
Fig. 4 – the fonts are too small. Why are data missing for S6 on the graph “A498/PAX2”? Why did the Authors not consider that depletion of S2 could activate this originally low-expressed gene, especially since such an effect was observed in other cell lines.
Figs. 5-7 – please enlarge the figures and fonts. Fig. 7 is not readable at all (bars & fonts & legends), especially when printed in black/white mode.
Also, Supplementary Figs require descriptions.
b/ The Authors did not comment on the condition of the siRNA-treated cells. Is there any chance that depletion of the studied PAX genes resulted in poor survival/induced apoptosis? Also, the only validation method is RT-qPCR. Western blot is more convincing. Frequently, lower expression of targeted genes does not always lead to protein depletion.
c/ It is not clear what is the purpose of the study presented in section 2.2. Why did the Authors select these lines and not others? Frequently, expression of various genes strongly depends on the type of tissue or even on the mutational status of the cells derived from the same type of cancer. This issue was not addressed. In my opinion, the Authors should better support the necessity for this set of experiments and address the obtained results in the Discussion.
3. The Discussion is short, but well-written. As stated above, it is missing comments on some of the performed analyses (2.2).
4. The Materials and Methods section - some minor typo errors occur. Please re-check the text.
Thank you very much.
Author Response
- The Introduction is well-written and informative.
We thank the reviewer for this comment.
- The Results section.
a/ First of all, the graphics are poor – in terms of quality, but also readability. They lack important information. E.g. Figure 2 A-C is missing information regarding gene names. Please provide the names of the genes on the Y-axis. Also, the description of the bars (e.g., “T”- tumor and “NT” non-tumor should be added).
We thank the reviewer for these comments. We have improved the quality and the readability of the graphics of the figures. On Figure 2 A-C we have added the gene names on the Y-axis, as suggested. We have revised Figure 2, so that this figure now only shows tumour-associated PAX gene expression.
In case of Fig. 3 – please provide the names of the cell lines (below the X-axis) on each graph.
In Figure 3, we have revised this figure to show only RCC cell lines, and we have provided the names of the cell lines below the X-axis on each graph, as suggested.
Fig. 4 – the fonts are too small. Why are data missing for S6 on the graph “A498/PAX2”? Why did the Authors not consider that depletion of S2 could activate this originally low-expressed gene, especially since such an effect was observed in other cell lines.
As stated for Reviewer #1, we have removed the previous Figure 4 from the manuscript.
Figs. 5-7 – please enlarge the figures and fonts. Fig. 7 is not readable at all (bars & fonts & legends), especially when printed in black/white mode.
We have enlarged the figures and fonts of the figures as suggested (which are now Figures 4-6).
Also, Supplementary Figs require descriptions.
We have included the descriptions together with the Supplementary Figures.
b/ The Authors did not comment on the condition of the siRNA-treated cells. Is there any chance that depletion of the studied PAX genes resulted in poor survival/induced apoptosis? Also, the only validation method is RT-qPCR. Western blot is more convincing. Frequently, lower expression of targeted genes does not always lead to protein depletion.
See our response below, after c/.
c/ It is not clear what is the purpose of the study presented in section 2.2. Why did the Authors select these lines and not others? Frequently, expression of various genes strongly depends on the type of tissue or even on the mutational status of the cells derived from the same type of cancer. This issue was not addressed. In my opinion, the Authors should better support the necessity for this set of experiments and address the obtained results in the Discussion.
In response to both b/ and c/, and as we have indicated to Reviewer #1, we have decided to remove the siRNA experiments from this paper, and accordingly Figure 4, and the associated text from section 2.2 have been deleted.
- The Discussion is short, but well-written. As stated above, it is missing comments on some of the performed analyses (2.2).
We have now expanded the Discussion and we have commented further on the analyses performed.
- The Materials and Methods section - some minor typo errors occur. Please re-check the text.
We have substantially revised multiple sections and figures in the manuscript, and we have attended to the typos and text in the Materials and Methods.
Round 2
Reviewer 1 Report
The authors have addressed my questions. The paper could be accepted.